# The Current Role of Stereotactic Body Radiation Therapy (SBRT) in Hepatocellular Carcinoma (HCC)

**DOI:** 10.3390/cancers14184383

**Published:** 2022-09-08

**Authors:** Tomoki Kimura, Toshiki Fujiwara, Tsubasa Kameoka, Yoshinori Adachi, Shinji Kariya

**Affiliations:** 1Department of Radiation Oncology, Kochi Medical School, Kochi University, Kohasu, Oko-cho, Nangoku-shi 783-8505, Kochi, Japan; 2Department of Radiation Oncology, Hiroshima Red Cross Hospital & Atomic-Bomb Survivors Hospital, 1-9-6 Sendamachi, Naka-ku, Hiroshima 730-8619, Hiroshima, Japan

**Keywords:** stereotactic body radiotherapy (SBRT), hepatocellular carcinoma (HCC), portal vein tumor thrombus (PVTT), inferior vena cava tumor thrombus (IVCTT), bridging therapy, oligometastasis

## Abstract

**Simple Summary:**

The role of stereotactic body radiotherapy (SBRT), which can deliver high radiation doses to focal tumors, has greatly increased in not only early-stage hepatocellular carcinoma (HCC), but also in portal vein or inferior vena cava thrombi, thus expanding this therapy to bridging transplantation and treating oligometastases from HCC in combination with immune checkpoint inhibitors. Major guidelines suggest that SBRT be regarded as a substitute therapy to conventional therapies, such as surgery, ablative therapies, transarterial chemoembolization, and systemic therapy. Further investigations are expected to establish the rationale for using SBRT in each situation.

**Abstract:**

The role of stereotactic body radiotherapy (SBRT), which can deliver high radiation doses to focal tumors, has greatly increased in not only early-stage hepatocellular carcinoma (HCC), but also in portal vein or inferior vena cava thrombi, thus expanding this therapy to pre-transplantation and the treatment of oligometastases from HCC in combination with immune checkpoint inhibitors (ICI). In early-stage HCC, many promising prospective results of SBRT have been reported, although SBRT is not usually indicated as a first treatment potion in localized HCC according to several guidelines. In the treatment of portal vein or inferior vena cava tumor thrombi, several reports using various dose-fraction schedules have shown relatively good response rates with low toxicities and improved survival due to the rapid advancements in systemic therapy. Although SBRT is regarded as a substitute therapy when conventional bridging therapies to transplantation, such as transarterial chemoembolization (TACE) and radiofrequency ablation (RFA), are not applicable or fail in controlling tumors, SBRT may offer advantages in patients with borderline liver function who may not tolerate TACE or RFA, according to several reports. For oligometastases, the combination of SBRT with ICI could potentially induce an abscopal effect in patients with HCC, which is expected to provide the rationale for SBRT in the treatment of oligometastatic disease in the near future.

## 1. Introduction

According to a recent report by the International Agency for Research on Cancer, among different types of cancer, liver cancer has the sixth highest incidence in the world, but its mortality is ranked the third highest [1]. Recently, both the incidence and mortality of liver cancer have decreased in many high-risk countries, including those in East- and Southeast Asia, because of vaccination against the hepatitis B virus (HBV) and the development of treatment against the hepatitis C virus (HCV) [1]. However, infections with viruses, such as HBV or HCV, are still major causes of cirrhosis and multifocal hepatocellular carcinoma (HCC); therefore, resection was selected as the initial treatment in only 38.3% of patients in Japan [2]. Recent technical advances in radiation therapy have made it possible to deliver high radiation doses to focal tumors; thus stereotactic body radiotherapy (SBRT) has been widely used in various tumors, such as lung, liver and bone tumors. Promising retrospective results of SBRT in early-stage HCC have reported that local control (LC) rates and overall survival (OS) rates have generally ranged from 66–100% and from 60–70% at 2–3 years, respectively [3,4,5,6]. However, due to a lack of specific evidence, SBRT should not be considered as a first-line treatment of localized HCC according to several guidelines [7,8,9]. The 2022 updated Barcelona Clinic Liver Cancer (BCLC) strategy showed that surgery, liver transplantation (LT), and ablative therapies, that is, radiofrequency ablation (RFA), are considered the first treatment options in early-stage HCC [7]. Although a randomized phase III study comparing SBRT with other modalities has not been reported until now, several prospective phase II studies have shown promising data [10,11,12,13,14,15,16,17].

The role of SBRT has become increasingly important not only in early-stage HCC, but also in HCC with portal vein or inferior vena cava thrombi, thus extending this therapy to bridging transplantation and treating oligometastases from HCC, such as bone and lung, in combination with immune checkpoint inhibitors (ICI). This article reviews the increasingly important roles of SBRT in HCC treatment.

## 2. SBRT in Early-Stage HCC

### 2.1. Treatment Outcomes

At this moment, SBRT could be regarded as a substitute therapy to surgery, LT, RFA, transarterial infusion chemotherapy or chemoembolization (TACE) [7,8,9]. Recent prospective studies showed comparable OS to the other modalities, almost 70% in 3 years, with an excellent LC, almost 90% in 2 or 3 years for small HCC [10,11,12,13,14,15,16,17]. Table 1 summarizes the prospective studies on SBRT in early-stage HCC. The toxicities of SBRT are relatively low, and the incidence of grade III or more toxicities ranged from 2% to 38% in Table 1. The most frequent adverse effects were associated with liver injury, such as the elevation of total bilirubin and transaminase and the decrease of platelets and ascites. Rim et al. reported a meta-analysis of 32 published studies involving 1950 HCC patients; OS and LC in 3 years were 48.3% and 83.9%, respectively, and grade III or more hepatic and gastrointestinal (GI) toxicities were 4.7% and 3.9%, respectively [18]. In these series, the Child–Pugh classification (CPC) was a significant prognostic factor for OS and toxicities [5,6,13,18]. Lasley et al. reported the results of SBRT in 38 patients with CPC-A and 21 patients with CPC-B; the 3-year OS was significant lower in patients with CPC-B (61% for CPC-A, 26% for CPC-B, *p* = 0.03). In addition, the incidence of grade III or more toxicities was higher in patients with CPC-B (11% for CPC-A, 38% for CPC-B); patients with CPC-B experiencing grade II or more liver toxicity had significantly higher dosimetric parameters of the normal liver [13]. GI toxicities have been reported [11]. Kang et al. reported that 5 (10.5%) of 47 patients experienced more than G3 GI toxicity, including grade IV gastric ulcer perforation in two patients (4.3%). They concluded that preexisting gastro-duodenal disease with cirrhosis was a significant risk factor, because in patients with liver cirrhosis, portal hypertension probably affects the gastrointestinal mucosal defensive and healing mechanisms, whereas liver cirrhosis increases GI toxicity. In general, it is recommended that the target proximity to the luminal GI tract should be more than 2 cm from the tumor. The incidence of central liver toxicities, such as central biliary tract (CBT) stenosis and portal vein (PV) thrombosis, are not so high [19,20,21]. Eriguchi et al. reported that only two patients (3.6% in 55 patients) experienced asymptomatic bile and concluded that SBRT for liver tumors adjacent to the CBT was feasible with minimal biliary toxicity [19]. However, Toesca et al. reported that grade III ≥ CBT stenoses were observed in seven patients (17.5% of 40 patients) [20] They recommended the limiting dose of CBT to be V_BED10_40 < 37 cc and V_BED10_30 < 45 cc. Takahashi et al. reported that grade III ≥ PV thrombi were observed in three patients (4.8% in 63 patients) [21]. They concluded that PV thrombosis may be needed to be considered in patients with a higher Child–Pugh class, with higher doses received to 2% of the PV volume.

SBRT is considered an alternative to other modalities, and most patients who undergo SBRT are non-naïve, therefore, previous treatments could affect the treatment results of SBRT, especially the OS in previous reports. Only a few reports have been published on SBRT in naïve patients [16,17,22]. Durand-Labrunie et al. reported on a prospective phase II study of SBRT using 45 Gy in 3 fractions in 43 naïve patients. The 2-year LC and OS were 98% and 69%, respectively, with 31% toxicities being greater than grade III [16]. Kimura et al. also reported on a prospective phase II study of SBRT using 40 Gy in 5 fractions in 36 naïve patients and showed that the OS and LC in 3 years were 78%, 90%, respectively, with 11% of toxicities being greater than grade III [17]. Considering that the eligible patients of these prospective studies were not suitable for resection, liver transplantation and RFA, the results of SBRT were comparable to those of the first treatment options in naïve patients [23,24].

**Table 1 cancers-14-04383-t001:** Prospective studies of SBRT in early-stage HCC.

Author/Year	Study Design	N	Median Tumor Size	BCLC * Stage C	Previous Treatment	Dose/Fraction (Gy/fr)	Prescription	Local Control	Overall Survival	Toxicity Grade 3≥
Andoliano, 2011, USA [10]	Phase I/II	60 (CPC-A/B ^#^: 36/24)	31 mm	17%	100%	42–60 Gy/3 fr	70–80% isodose	94.6% (2y)	68.7% (2y)	10.7%
Kang, 2012, Korea [11]	Phase II	47 (CPC-A/B: 41/6)	29 mm	N.A. **	N.A. **	24–48 Gy/3 fr	80% isodose	90% (2y)	67% (2y)	25%
Bujold, 2013, Canada [12]	Phase I/II	102 (CPC-A/B: 102/0)	72 mm	65.7%	52%	24- 54 Gy/6 fr	N.A. **	87.0% (1y)	34.0% (2y)	30%
Lasley, 2015, USA [13]	Phase II	CPC-A: 38	N.A.	N.A.	N.A.	48 Gy/3 fr	80–90% isodose	91% (3y)	61% (3y)	11%
		CPC-B: 21	N.A.	N.A.	N.A.	40 Gy/5 fr	80–90% isodose	82% (3y)	26% (3y)	38%
Takeda, 2016, Japan [14]	Phase II	90 (CPC-A/B: 82/8)	23 mm	16%	64%	40 or 35 Gy/5 fr	60–80% isodose	96.3% (3y)	66.7% (3y)	15%
Jang, 2020, Korea [15]	Phase II	65 (CPC-A/B: 64/1)	24 mm	6.2%	100%	42–60 Gy/3 fr	90% isodose	95% (3y)	76% (3y)	2%
Durand-Labrunie, 2020, France [16]	Phase II	43 (CPC-A/B: 37/6)	28 mm	0%	0%	45 Gy/3 fr	80% isodose	94% (2y)	69% (2y)	31%
Kimura, 2021, Japan [17]	Phase II	36 (CPC-A/B: 33/3)	23 mm	0%	0%	40 Gy/5 fr	70% isodose	90% (3y)	78% (3y)	11%

Abbreviations: * BCLC: Barcelona Clinic Liver Cancer, ** N.A.: not available, ^#^ CPC-A/B: Child-Pugh class A/B.

There are several unresolved issues in SBRT in early-stage HCC. The first issue is the timing of response evaluation. The evaluation method for SBRT is usually judged by whether there is early arterial enhancement of the tumor using dynamic enhanced CT or MRI [25,26]. According to the Modified Response Evaluation Criteria in Solid Tumors (mRECIST), a complete response (CR) was defined as the “Disappearance of any intratumoral arterial enhancement in all target lesions” [25]. However, this intratumoral arterial enhancement may be prolonged over 6 or more months in several cases, and this fact confuses the timing of response evaluation. In a previous report, Kimura et al. evaluated the patterns of dynamic enhanced CT appearance of tumor responses after the completions of SBRT. They observed residual early arterial enhancement in 19 lesions (28.4%) more than 3 months after SBRT in 59 patients with 67 tumors [27]. Based on this result, we propose that the response evaluation at 6 months, not at 3 months, after the completion of SBRT is the appropriate time point, because in most cases there was ab observed disappearance of residual early arterial enhancement within 6 months. Figure 1 shows a typical case of CR at 3.5 months after the completion of SBRT. The second issue is the optimal dose-fraction schedule. The several dose-fraction schedules are shown in Table 1. Kim et al. analyzed the dose–response relationship in a multi-institutional retrospective cohort that included 510 patients treated with SBRT [28]. Patients treated with a biological effective dose (BED) ≥ 100 Gy showed a better 2-year freedom from local progression (FFLP) and OS than did patients treated with a BED < 100 Gy (FFLP, 89% vs. 69%; OS, 80% vs. 67%; *p* < 0.001). In addition, a multivariate analysis before and after propensity score-matching (PSM) in 198 selected patients between BED ≥ 100 Gy and BED < 100 Gy, identified BED ≥ 100 Gy as the main prognostic factor for both FFLP and OS (*p* < 0.01). Higher dose-fraction schedules may improve LC and OS. The third issue is the combination with TACE. Kimura et al. compared SBRT alone (28 patients) with SBRT+TACE (122 patients) in small HCC, the median tumor size was < 20 mm, retrospectively [29]. The 2 year OS and local progression-free survivals (LPFS) for SBRT alone and SBRT+TACE groups were 78.6% and 80.3% (*p* = 0.6583) and 71.4% and 80.8% (*p* = 0.9661), respectively. On the other hand, Su et al. also compared SBRT alone (50 patients) with SBRT+TACE (77 patients) in large HCC, with median tumor size being 85 mm, retrospectively [30]. The 5 year OS was significantly higher in the SBRT + TAE/TACE group (46.9%) than that in the SBRT alone group (32.9%; *p* = 0.049). The LPFS did not differ significantly between the two groups. These opposite results suggested that the combination of SBRT and TACE has the potential to improve treatment results, compared with SBRT alone, especially in patients with larger HCC, such as those of >5 cm, but SBRT alone could be a significant treatment option for patients with small HCC, such as those of <2 cm. To resolve these issues, further prospective studies are warranted.

### 2.2. Comparison to the Other Treatments

Recently, several PSM studies, which compared SBRT with the different modalities of local therapy, have been published. Table 2 summarizes these PSM studies in early-stage HCC.

TACE is considered an alternative to resection or RFA, and its eligibility is regardless of tumor conditions, such as its size, number and location in several guidelines [7,8,9]. Spair et al. reported the comparison between TACE in 84 patients with 114 HCCs and SBRT in 125 patients with 173 HCCs using PSM [31]. SBRT produced significantly better LC (2 year: 91% for SBRT and 23% for TACE, *p* < 0.001) with lower grade III (and lower) toxicity (8% for SBRT and 13% for TACE, *p* = 0.05) despite no difference in OS. A meta-analysis, which compared TACE alone with radiation therapy (RT) including SBRT with TACE, showed that the median survival for TACE with RT (22.7 months) was significantly better than that for TACE alone (13.5 months) (*p* < 0.001) [32].

Several comparative PSM studies with RFA, which is a first-line treatment, have been reported [33,34,35,36]. Recent large-size studies showed significantly better LC for SBRT compared to RFA and comparable OS and toxicities [35,36]. On the other hand, Rajyaguru et al. showed opposite outcomes with RFA producing significantly better OS; the 5 year OS of RFA was 29.8% and that of SBRT was 19.3% (*p* = 0.001) using the National Cancer Database [34]. However, several limitations, such as selection bias, have been pointed out by several investigators. Several meta-analyses, which compared SBRT with RFA, showed that the LC of SBRT was better or comparable to that of RFA, especially for a tumor size of ≥ 2 cm, but the opinions were divided regarding OS [37,38,39]. Pan et al. concluded that OS of SBRT was inferior to that of RFA because of the tumor burden or liver profiles of the enrolled study participants [37]; in contrast, Wang et al. concluded that SBRT was well-tolerated with an OS equivalent to that with RFA [39]. These findings could suggest the limitations of these retrospective studies, therefore further investigations are still needed.

**Table 2 cancers-14-04383-t002:** Comparison with the other modalities.

Author/Year	Study Design	Modality	N (Matched)	Median Tumor Size	Local Control	*p*-Value	PFS	*p*-Value	Overall Survival	*p*-Value	Toxicity Grade 3≥	*p*-Value
Spair, 2018, USA [31]	IPTW *	SBRT	125	23 mm	91% (2y)	0.008	26.9% (2y) ^#^	<0.001	54.9% (2y)	0.21	8%	0.05
		TACE	84	29 mm	23% (2y)		10.7% (2y) ^#^		34.9% (2y)		13%	
Wahl, 2016, USA [33]	IPTW	SBRT	63	22 mm	N.A.	-	83.8 % (2y) ^#^	N.S. ^†^	52.9% (2y)	N.S.	5%	0.31
		RFA	161	18 mm	N.A.		80.2 (2y) ^#^		46.3% (2y)		11%	
Rajyaguru, 2018, USA [34]	PSM **	SBRT	296 (275)	N.A. ^¶^	N.A.	-	N.A.	-	19.3% (5y)	<0.001	N.A.	-
		RFA	3684 (521)	N.A.	N.A.		N.A.		29.8% (5y)		N.A.	
Hara, 2019, Japan [35]	PSM	SBRT	143 (106)	18 mm	93.6% (3y)	<0.001	N.A.	-	69.1% (3y)	0.86	0 %	N.A.
		RFA	231 (106)	17 mm	79.8% (3y)		N.A.		70.4% (3y)		2%	
Kim, 2020, Korea [36]	PSM	SBRT	496 (313)	21 mm	80.6% (2y)	<0.001	N.A.	-	77.6% (2y)	0.308	1.6%	0.268
		RFA	1568 (313)	22 mm	76.3% (2y)		N.A.		71.1 (2y)		2.6%	
Su, 2017, China [40]	PSM	SBRT	82 (33)	33 mm	N.A.	-	43.9% (5y)	0.945	74.3% (5y)	0.45	N.A.	-
		Surgery	35 (33)	35 mm	N.A.		35.9% (5y)		69.2% (5y)		N.A.	
Nakano, 2018, Japan [41]	PSM	SBRT	27 (27)	18.4 mm	N.A.	-	16.4% (5y)	0.0512	47.8% (5y)	0.0149	3.7%	N.A.
		Surgery	254 (54)	17.6 mm	N.A.		33.8% (5y)		75.2% (5y)		9.1%	
Sun, 2020, China [42]	PSM	SBRT	122 (104)	26 mm	N.A.	-	49% (5y)	0.350	71% (5y)	0.673	0	N.A.
		Surgery	195 (104)	27 mm	N.A.		47.3% (5y)		70.7% (5y)		21.5%	

Abbreviations: * IPTW: inverse probability of treatment weighting to the Kaplan–Meier method and Cox models, ** PSM: propensity score-matching. ^#^ FFLP: Freedom from in-liver (local) progression. ^¶^ N.A.: not available, ^†^ N.S.: not significant.

In comparisons with resection, which is also a first-line treatment, PSM studies from China showed that SBRT produced a similar OS to that in resection [40,42]; in addition, Su et al. showed that an advantage of SBRT over resection was its lesser degree of invasiveness [40]. In contrast, Nakano et al. from Japan reported that the 5 year OS and PFS rates for the resection and SBRT groups were 75.2% vs. 47.8% (*p* = 0.0149) and 33.8% vs. 16.4% (*p* = 0.0512), respectively, and a multivariate analysis showed that resection was a significant favorable factor for OS and PFS [41]. They concluded that resection should be considered a first treatment option for potentially resectable patients.

Charged particle therapy (CPT), including carbon ion therapy and proton beam therapy (PBT), provides physical and biological advantages compared with SBRT, which uses a photon beam. In a physical aspect, the “Bragg peak”, which is a rapid energy fall-off at a specific depth, allows for the delivery of a very localized dose distribution that potentially reduces the incidence of hepatic toxicity. In a biological aspect, the greater relative biological effectiveness of CPT, especially carbon ion therapy, compared with SBRT, could be expected to improve LC and OS. Based on this background, CPT is promising for large tumors with relatively less toxicity. Qi et al. reported a meta-analysis of the CPT and SBRT in patients with HCC [43]. The OS for CPT was similar to that for SBRT, and the toxicity tended to be lower for CPT compared to that for SBRT. From 2022, CPT for HCC (≤4 cm) is covered by the national health insurance in Japan. Now, a prospective non-randomized trial on PBT vs. surgery for operable untreated HCC is ongoing (Japanese Clinical Oncology Group, JCOG1305).

Further studies, including randomized phase III studies to define which patients are more suitable for each curative local treatment, are needed.

### 2.3. Repeated SBRT

According to the latest Japanese survey, recurrence was reported within two years of diagnosis in 50.5% patients with HCC [2]. Because of the multifocal nature, intrahepatic recurrence is the one most frequently observed in 80–95% of cases [44], and Imamura et al. reported that two types of recurrence may be distinguished: early and late recurrences [45]. Early recurrence is considered a metastatic occurrence and late recurrence a multicentric occurrence of HCC, therefore, the late recurrence shares the same risk factors as primary HCC [45]. Considering these recurrent patterns and their frequencies, repeated locoregional therapies play an important role. The treatment strategy for patients with intra-hepatic recurrent HCC after initial treatments, such as surgery or RFA, should principally follow the same eligibility which was used for naïve HCC patients [46]. SBRT could also be considered a substitute therapy to surgery or RFA in this situation. Kimura et al. reported the results of repeated SBRT in 81 patients with 189 tumors of two courses or more (median two times; ranged from two to four times) [47]. The 5 year local recurrence rate, OS and liver-related death rates from the first SBRT were 6.3%, 60.4% and 32.9%, respectively. The 3 year OS and liver-related death rates from the second SBRT were 61.0% and 34.5%, respectively, with almost the same frequency of grade III toxicity between the first and second SBRT (first: 11%; second: 15%, *p* = 0.48). Repeated SBRT for patients with intra-hepatic recurrent HCC achieved satisfactory LC and OS without severe toxicities; therefore, SBRT could be a good treatment option in these recurrent cases.

## 3. SBRT for Portal Vein or Inferior Vena Cava Tumor Thrombi

Macroscopic vascular invasions (MVIs), such as those in the portal vein, hepatic vein and inferior vena cava, are observed frequently at the first diagnosis of HCC. In the latest Japanese nationwide survey, vascular invasion was observed in the portal vein in 13.2%, hepatic veins in 6.2%, and bile duct in 3.4% of patients [2]. BCLC staging defines patients with portal vein invasion as advanced stage (C) and recommended systemic therapy [7]. Sorafenib has been selected as the first-line systemic therapy, but a recent phase III study on patients with unresectable HCC showed that the combination of atezolizumab and bevacizumab resulted in better OS and PFS than did sorafenib (1-year OS 67.2% vs. 54.6%) [48]. As rapid advances in systemic therapy take place, what is the role of radiation therapy for patients with MVI? First, radiation therapy is considered to prevent symptoms such as bleeding from esophageal varices, secondary Budd–Chiari syndrome and pulmonary tumor thrombi [49]. Second, radiation therapy is considered to improve survival because portal vein tumor thrombi (PVTT) or inferior vena cava tumor thrombi (IVCTT) are associated with widespread intrahepatic and extrahepatic dissemination by the spread of tumor cells through the portal tract [50,51,52].

Although three-dimensional radiation therapy (3D-CRT) has been used recently with TACE, and hepatic arterial infusion chemotherapy (HAIC) and molecular targeted drugs for HCC with MVI, SBRT has often been applied instead of 3D-CRT due to achievement of higher BED doses within a shorter duration of treatment. Table 3 summarizes several studies on SBRT in advanced HCC with PVTT or IVCTT [53,54,55,56,57,58,59,60,61,62]. Shuqun et al. classified the extent of PVTT into four types: Type I: tumor thrombus involving segmental or sectoral branches of the portal vein or above; Type II: tumor thrombus involving the right/left portal vein; Type III: tumor thrombus involving the main portal vein; Type IV: tumor thrombus involving the superior mesenteric vein (Cheng’s classification) [63]. They also reported the median survival periods for patients of groups I (*n* = 17), II (*n* = 26), III (*n* = 35) and IV (*n* = 6) as 10.1, 7.2, 5.7 and 3.0 months, respectively (*p* = 0.0001) and concluded that these types of tumor thrombi could be prognostic factors for HCC patients with PVTT. Shui et al. suggested the basic criteria for applying SBRT as follows: (1) tumor thrombus involving the main trunk and/or first branches of the portal vein (most of patients were classified as Type II-III in Cheng’s classification), unsuitable for surgery or TACE; (2) PS(ECOG) 0–2; (3) No refractory ascites; (4) CPC A and B, or class C with good PS; (5) No previous radiotherapy to the liver; (6) More than 700 cc of uninvolved liver [58]. In Table 3, various dose-fractions were used, such as 36–50 Gy in 3–15 fractions; however, the optimal dose-fraction schedule is still unknown. Li et al. compared the OS, PFS and LC of SBRT using a BED assumed at an α/β ratio of 10 (BED10) ≥ 100 Gy with those of SBRT using a BED10 < 100 Gy in HCCs with PVTT, and reported that those of the BED10 ≥ 100 Gy group were significantly improved [61]. They concluded that SBRT using a BED10 ≥ 100 Gy is recommended, if dose constraints are kept. Rim et al. reported on a meta-analysis that compared 3D-CRT and SBRT for HCC with PVTT [64]. Although OS did not differ among the two modalities, response rate and grade III and above complications were better in SBRT.

**Table 3 cancers-14-04383-t003:** SBRT for portal vein/inferior vena cava tumor thrombus.

Author/Year	Study Design	N	Techniques	Total Dose (Range)	Fractions	Response Rate	1-Year OS (Median)	Toxicity
Tse, 2008, USA [53]	Phase I	16 (total 41)	Static IMRT	36 Gy (24–54 Gy)	6 fr	25% (CR 6%, PR 19%)	48% (11.6 m)	23% (CP class *, all)
Choi, 2008, Korea [54]	retrospective	9 (total 31)	CyberKnife	36 Gy (30–36 Gy)	3 fr	44.4% (CR 11.1%, PR 33.3%)	43.2% (8 m)	16.1% (CP class, all)
Xi, 2013, China [55]	retrospective	41	VMAT	36 Gy (30–48 Gy)	6 fr	75.6% (CR 36.6%, PR 39%)	50.3% (13 m)	2.4% (grade 3≥) **
Kang, 2014, China [56]	retrospective	101	Static IMRT	40.2 Gy (21–60 Gy)	6 fr	70.3% (CR 18%, PR 53%)	50–58.8% (12–15 m)	34.7% (CP class)
Matsuo, 2016, Japan [57]	retrospective	43	CyberKnife/TrueBeam	50 Gy (36–55 Gy)	10–15 fr	67% (CR 1%, PR 65%)	49.3% (11 m)	8.3% (CP class)
Shui, 2018, China [58]	retrospective	70	VMAT	40 Gy (25–50 Gy)	5 fr	77.4% (CR 0%, PR 77.4%)	40% (10 m)	4.3% (CP class)
Choi, 2020, Korea [59]	prospective	24	CyberKnife	45 Gy (39–45 Gy)	3–4 fr	54.2% (CR 8.3%, PR 45.8%)	67.5% (20.8 m)	25% (grade 3≥) ^#^
Que, 2020, Taiwan [60]	retrospective	36	CyberKnife alone	40 Gy (36–40 Gy)	3–5 fr	75% (CR 25%, PR 50%)	33.3% (7 m)	8.3% (grade 3≥) ^†^
		18	CyberKnife + sorafenib			77.7% (CR 33.3%, PR 44.4%)	55.6% (12.5 m)	27.8% (grade 3≥) ^††^
Li, 2021, China [61]	retrospective	102	CyberKnife	42 Gy (30–50 Gy)	3–5 fr	62.1% (2-year local control)	46.5% (10 m)	11.8% (CP class)
Munoz-Schuffeneger [62], 2021, Canada	retrospective	128	Linac	33.3 y (27–54 Gy)	5 fr	87.4% (1-year local control)	N.A. (18.3 m)	27.6% (CP class)

Abbreviations: * CP class: progression of Child–Pugh class from A to B or C, ** Evaluated by CTCAE ver3.0, ^#^ Evaluated by CTCAE ver4.0. ^†^ Liver enzyme adverse effects evaluated by CTCAE ver4.0, ^††^ leukopenia or thrombocytopenia evaluated by CTCAE ver4.0.

Most of the studies were retrospective; further prospective studies on SBRT with or without systemic therapies are needed in order to establish the role of SBRT. Now, a prospective randomized trial of sorafenib versus SBRT followed by sorafenib in HCC is ongoing (Radiation Therapy Oncology Group, RTOG1112).

## 4. SBRT for a Bridging Therapy to Liver Transplantation (LT)

According to the BCLC strategy, LT is the standard therapy for patients who meet the Milan criteria, which comprise single tumors of less than 5 cm or three or fewer nodules of 3 cm or less [7]. SBRT could be considered bridging therapies of LT as locoregional therapies. Table 4 summarizes several studies on SBRT as bridging therapy [65,66,67,68,69,70,71,72]. Because it would be difficult to undertake prospective studies, all studies were retrospective, and the numbers of patients were limited. OS ranged from 75 to 100%, disease-free survival was almost 75% and the drop-off rate ranged from 0% to 33% with various dose-fraction schedules.

**Table 4 cancers-14-04383-t004:** Results of stereotactic body radiotherapy as a bridging therapy.

Author/Year	Study Design	N	Tumor Size	Child-Pugh A	Total Dose/Fractions	Median Months to Transplantation	Disease-Free Survival	Overall Survival	Pathological CR Rate	Drop-Off Rate
Sandroussi, 2010, Canada [65]	retrospective	10	79 cc	40%	33 Gy/6 fr	5	N.A. *	N.A.	N.A.	20% (8/10)
O’Connar, 2012, USA [66]	retrospective	10	34 mm	80%	51 Gy/3 fr	3.5	N.A.	100% (5y)	27%	0%
Katz, 2012, USA [67]	retrospective	18	40 mm	16.7%	50 Gy/10 fr	6	N.A.	N.A.	18.2%	33% (12/18)
Barry, 2016, Canada [68]	retrospective	38	60.5 cc	42%	36 Gy/6 fr	N.A.	79% (5y)	76% (5y)	N.A.	13% (5/38)
Mannina, 2017, USA [69]	retrospective	38	24 mm	45%	40 Gy/5 fr	8.1	74% (3y)	77% (3y)	23.5%	N.A.
Moore, 2017, Israel [70]	retrospective	16	25 mm	45.5%	54 Gy/3 fr	4.8	N.A.	N.A.	27.3%	31.3% (11/16)
Sapisochin, 2017, Canada [71]	retrospective	36	45 mm	61%	36 Gy/6 fr	13.7	74% (5y)	75% (5y)	13.3%	16.7% (30/36)
Wang, 2021, Taiwan [72]	retrospective	14	44.5 mm	78.6%	45 Gy/5 fr	8.4	18.3 months (median)	37.8 months (median)	23.1%	0%

Toxicity ≥ Grade 3 after SBRT was 0% in all studies. Abbreviation: * N.A.: not available.

In these studies, Sapisochin et al. compared the robust intention-to-treat analysis with several modalities; 379 patients were divided into three bridging treatment groups, such as SBRT (36 patients), TACE (99 patients) and RFA (244 patients) [71]. The 5 year OS after LT was very similar (*p* = 0.7) among the SBRT groups (75%), the TACE group (69%) and the RFA group (73%). In addition, the rates of drop-out were also similar (*p* = 0.7) among the SBRT groups (16.7%), the TACE group (20.2%) and the RFA group (16.8%). The rate of impaired liver function was significantly higher (*p* = 0.001) in the SBRT group (38.9%) than in the TACE (19.4%) and RFA groups (13%) because of selection bias, such as poor liver function and outside Milan criteria in the SBRT group. SBRT may be safely utilized as a bridge to LT in patients with HCC or as an alternative when conventional bridging therapies, such as TACE, and RFA, are not applicable, or fail, in controlling tumors. SBRT may offer advantages in patients with borderline liver function who may not tolerate TACE or RFA.

## 5. SBRT for Extrahepatic Metastasis

The most common sites of extrahepatic metastasis were the lungs, bone, lymph nodes, adrenal glands, and brain, with an incidence of 34.7% in the lungs, 29.3% in the lymph nodes and 17.2% in the bone, according to the latest Japanese survey [2]. The BCLC staging system defines patients with extrahepatic metastasis (often painful bone metastases) as advanced stage (C) and recommends systemic therapy [7]. The role of radiation therapy in this stage is usually palliative care. However, recently the concept of oligometastasis, which is an intermediate state between localized and systemically metastasized disease, has been expanded [73]. Several studies have reported that the use of aggressive local therapies in oligometastatic disease has increased survival in patients with various cancers, such as lung, colorectal and prostate cancers [73,74,75,76]. SBRT is frequently used in oligometastatic disease as a local therapy due to its lesser invasiveness. Kim et al. showed that the level of soluble programmed cell death ligand-1 was significantly increased in patients with HCC who were treated with SBRT [77]. This finding suggests that the combination of SBRT with ICI could potentially induce an abscopal effect also in patients with HCC. Further investigations are expected to establish the evidence for SBRT in oligometastatic disease in the near future.

## 6. Discussion—Future Perspectives

As described above, the current role of SBRT in HCC has been greatly expanded, and SBRT has become an essential modality in several aspects of HCC treatment. Considering the future perspectives for SBRT, there are several issues that should be resolved, such as an establishment of the evidence for early-stage HCC, especially in naïve patients, the combination of SBRT and ICI in advanced HCC, and the segregation of SBRT and CPT. First, SBRT in early-stage HCC; as shown in Table 1, the results of many prospective studies on SBRT in early-stage HCC are promising. However, the levels of evidence in external beam radiation therapy, including SBRT, have been low because of a lack of phase III studies [78]. In addition, the comparison to the other treatments, shown in Table 2, does not also include Phase III studies. It would be very difficult to compare different modalities in a randomized trial. For example, the SURF trial, which is a randomized controlled trial of surgery versus RFA for small HCC, showed that the target number of patients (600 patients) was not reached because the speed of patient accrual was low [79]. On the other hand, Kim et al. reported the results of a randomized phase III trial, comparing PBT and RFA for patients with recurrent or residual HCC [80]. There were several limitations in this study. The eligibility criteria of this study were patients with recurrent HCC, not naïve HCC. In addition, the primary endpoint was a 2 year local progression-free survival (LPFS), which was defined as the time from the commencement date of each intervention to the date of local progression, rather than progression-free survival or OS. However, PBT showed LPFS values that were non-inferior to those for RFA. The number of patients enrolled was 144, which is not so large, but the possibility of a Phase III study with different modalities would be expected. In fact, Phase III trials of the SBRT vs. RFA for patients with untreated HCC are currently being conducted in Asian countries, as follows, and the results are awaited.

NCT03898921 (ClinicalTrials.gov): Radiofrequency Ablation (RFA) Versus Stereotactic Body Radiotherapy (SBRT) for Small hepatocellular Carcinoma: A Phase III, Prospective, Randomized, Open, Parallel Controlled Clinical Trial. Primary endpoint: OS, The number pf register: 270 patients, China.

NCT05433701 (ClinicalTrials.gov): A Phase III Randomized Controlled Non-inferiority Trial to Compare SBRT vs. RFA for Unresectable, Small (≤3 cm) HCC. Primary endpoint: LPFS, The number pf register: 162 patients, Korea.

Second, the combination of SBRT and ICI for extrahepatic metastasis, which has been supported by basic theories: Sharabi et al. described that SBRT could produce immune-mediated systemic responses and induce an “abscopal effect”; therefore, the combination of SBRT and ICI increases tumor cell’s susceptibility to immune-mediated cell death [81]. In a clinical setting, Tang et al. reported a Phase I trial testing SBRT with cytotoxic T lymphocyte antigen 4 (CTLA-4) and ipilimumab for patients with metastatic solid tumors of the liver or lung that were refractory to standard therapies. They concluded that combining SBRT and ipilimumab was safe with a 10% partial response in non-irradiated lesions, and irradiation to the liver produced greater T-cell activation than did irradiation to the lung [82]. From these results, the combination of SBRT and ICI could improve survival more than SBRT alone for patients with oligometastatic disease. Table 5 shows the summary of ongoing prospective trials of immunotherapy combined with SBRT for HCC.

Third, the segregation of SBRT and CPT has been an important issue. As described in the previous part, there is no doubt that CPT is advantageous due to its physical aspects, especially for large size HCC. On the other hand, for early-stage HCC, although CPT is advantageous from the viewpoint of adverse events from the meta-analysis, LC has no large difference when compared to SBRT at present. Considering accessibility and cost, it seems realistic at present to segregate large HCCs (>4–5 cm) that are difficult to control with SBRT. The NCCN guidelines state that “proton beam therapy may be appropriate in specific situations” [8]. To establish evidence, several studies comparing it with other local modalities, such as RFA, are ongoing; there is a possibility that CPT has a higher priority than other modalities in the selection of local treatment for HCC in the near future.

## 7. Conclusions

The current role of SBRT in HCC has been expanded to include patients with not only early-stage HCC, but also in PVTT/IVCTT, bridging transplantations and oligometastases. Further investigations are expected to establish the rationale for using SBRT in each situation.

## Figures and Tables

**Figure 1 cancers-14-04383-f001:**
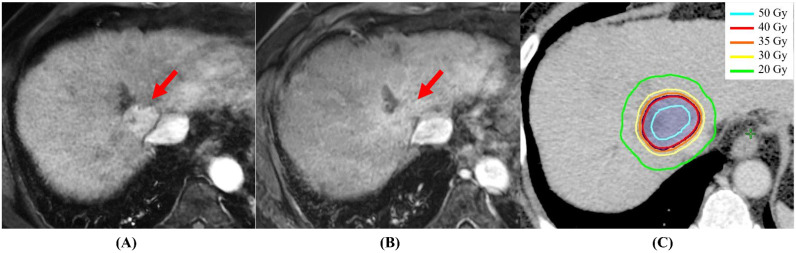
A typical case of complete response at 3.5 months after the completion of SBRT. (**A**) Dynamic MRI appearance (arterial phase) before SBRT; the early arterial enhancement is obvious (red arrow). (**B**) Dynamic MRI appearance (arterial phase) after 3.5 months; the early arterial enhancement has disappeared (red arrow). (**C**) Dose distribution of SBRT: the prescribed dose of 40 Gy covered 95% of PTV with 125% maximum dose of 40 Gy (80% isodose) in 4 fractions.

**Table 5 cancers-14-04383-t005:** Summary of ongoing prospective trials of immunotherapy combined with SBRT for advanced HCC. (ClinicalTrials.gov, accessed on 14 July 2022).

Trials (Country)	Type of Disease	Design	Number of Pts	Interventions	Primary Endpoint
NCT04547452 (China)	Stage IV HCC (Liver or lung or any metastatic lesion)	Randomized Phase II	42: SBRT+PD-1 ^#^ 42: PD-1 alone	RT: SBRT Drug: Sintilimab (PD-1)	24-week progression-free survival rate
NCT 05396937 (China)	Metastatic HCC (extrahepatic dissemination)	Phase II	42	RT: SBRT Drug: Atezolizumab, Bevacizumab	Objective response rate
NCT04988945 (China)	Tumor size 5–25 cm and number of lesions ≤3	Phase II	33	Procedure: TACE RT: SBRT Drug: Durvalumab, Tremelimumab	Downstaging for hepatectomy
NCT03817736 (China)	Tumor size 5–15 cm or number of lesions ≤3 or segmental portal vein involvement	Phase II	33	Procedure: TACE RT: SBRT Drug: ICI ^†^	Number of Patients Amendable to Curative Surgical Interventions
NCT04857684 (USA)	Resectable HCC	Phase I	20	RT: SBRT Drug: Atezolizumab, Bevacizumab	Proportion of patients with grade 3–4 treatment-related adverse events as assessed by CTCAE v5.0
NCT04913480 (China)	Stage C or earlier HCC based on BCLC * staging	Phase II	37	RT: SBRT Drug: Durvalumab	Progression-free survival at 1 year
NCT05185531 (China)	Medically fit to undergo surgery as determined by the treating medical and surgical oncology team	Phase I	20	RT: SBRT Drug: Tislelizumab (PD-1)	Delay to surgery overall response rate after neoadjuvant SBRT + Tislelizumab
NCT03316872 (Canada)	maximum 10 lesions to be treated, and total tumor diameter to be treated <20 cm	Phase II	30	RT: SBRT Drug: Pembrolizumab	Overall response rate

Abbreviations: * BCLC: Barcelona Clinic Liver Cancer, ^#^ PD-1: Programmed death receptor-1, ^†^ ICI: immune checkpoint inhibitors.

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
