# Peer review of "The Current Role of Stereotactic Body Radiation Therapy (SBRT) in Hepatocellular Carcinoma (HCC)"

_cancers, 2022, doi:10.3390/cancers14184383_

Round 1

Reviewer 1 Report

Don't cut tables between pages.

Table 2 needs to be reformated.  The headings need to be respaced

Figure 1C would look better is the beam was removed and only the isodose lines showed.  It would also be good to include a legend showing color and associated dose 

table 3 is incomplete and needs reformating

As one of the advantages of SBRT is that it is non-invasive and well tolerated.  It could be helpful to elaborate on the toxicity profile of SBRT for HCC, so the the readers understand the limited risks and side effects and have some numbers to reference in regard to how common each of the various side effects is.  

Author Response

Answers to the reviewers 1’ comments and suggestions

Comments and Suggestions for Authors

Don't cut tables between pages.

Answer

Thank you for your comments.

I modified.

Table 2 needs to be reformated. The headings need to be respaced.

Answer

Thank you for your comments. I reformatted Table 2.

Figure 1C would look better is the beam was removed and only the isodose lines showed. It would also be good to include a legend showing color and associated dose.

Answer

Thank you for your comments. I modified Figure 1C to remove the beam and add the associated dose.

Table 3 is incomplete and needs reformatting.

Answer

Thank you for your comments. I reformatted Table 3.

As one of the advantages of SBRT is that it is non-invasive and well tolerated. It could be

helpful to elaborate on the toxicity profile of SBRT for HCC, so the the readers understand the

limited risks and side effects and have some numbers to reference in regard to how common

each of the various side effects is.

Answer

Thank you for your comments.

I added the following sentences about toxicities of SBRT in “2. SBRT in Early-Stage HCC” part to be useful for the readers of this journal (Red character).

“2.1. Treatment Outcomes

At this moment, SBRT could be regarded as a substitute therapy to surgery, LT, RFA, transarterial infusion chemotherapy or chemoembolization (TACE) [7-9]. Recent prospective studies showed comparable OS to the other modalities, almost 70% in 3 years, with excellent LC, almost 90% in 2 or 3 years for small HCC [10-17]. Table 1 summarizes the prospective studies on SBRT in early-stage HCC. The toxicities of SBRT are relatively low, the incidence of grade 3 or more toxicities ranged from 2% to 38% in Table 1. The most frequent adverse effects were associated with liver injury, such as the elevation of total bilirubin and transaminase and the decrease of platelets and ascites. Rim et al. reported a meta-analysis of 32 published studies involving 1950 HCC patients, OS and LC in 3 years were 48.3% and 83.9%, respectively, and grade 3 or more hepatic and gastrointestinal (GI) toxicities were 4.7 % and 3.9 %, respectively [18]. In these series, Child-Pugh classification (CPC) was a significant prognostic factor for OS and toxicities [5, 6, 13, 18]. Lasley et al. reported the results of SBRT in 38 patients with CPC-A and 21 patients with CPC-B; the 3-year OS was significant lower in patients with CPC-B (61% for CPC-A, 26% for CPC-B, p=0.03). In addition, grade 3 or more toxicities was higher in patients with CPC-B (11% for CPC-A, 38% for CPC-B), patients with CPC-B experiencing grade 3 or more liver toxicity had significantly higher dosimetric parameters of the normal liver [13]. GI toxicities have been reported [11]. Kang et al. reported that 5 (10.5%) of 47 patients experienced more than G3 GI toxicity, including grade 4 gastric ulcer perforation in 2 patients (4.3%). They concluded that preexisting gastro-duodenal disease with cirrhosis was a significant risk factor, because in patients with liver cirrhosis, portal hypertension probably affects the gastrointestinal mucosal defensive and healing mechanisms, whereas liver cirrhosis increases GI toxicity. In general, it is recommended that the target proximity to the luminal GI tract should be more than 2 cm from the tumor. The incidence of central liver toxicities, such as central biliary tract (CBT) stenosis and portal vein (PV) thrombosis, are not so high [19-21]. Eriguchi et al. reported that only 2 patients (3.6% in 55 patients) experienced asymptomatic bile and concluded that SBRT for liver tumors adjacent to the CBT was feasible with minimal biliary toxicity [19]. However, Toesca et al. reported that Grade 3 ≥ CBT stenoses were observed in 7 patients (17.5% of 40 patients) [20] They recommended the limiting dose of CBT to VBED1040 < 37 cc and VBED1030 < 45 cc. Takahashi et al. reported that Grade 3 ≥ PV thrombi were observed in 3 patients (4.8% in 63 patients) [21] They concluded that PV thrombosis may be necessary to be considered in patients with a higher Child-Pugh class, with higher doses received to 2% of the PV volume.”

I also added the following references in “Reference” part.

References

  1. Eriguchi T, Takeda A, Sanuki N, Oku Y, Aoki Y, Shigematsu N, Kunieda E. Acceptable toxicity after stereotactic body radiation therapy for liver tumors adjacent to the central biliary system. Int J Radiat Oncol Biol Phys. 2013;85:1006–
  2. Toesca DAS, Osmundson EC, Eyben RV, Shaffer JL, Lu P, Koong AC, Chang DT. Central liver toxicity after SBRT: an expanded analysis and predictive nomogram. Radiother Oncol. 2010;76:130–
  3. Takahashi S, Kimura T, Kenjo M, Nishibuchi I, Takahashi I, Takeuchi Y, Doi Y, Kaneyasu Y, Murakami Y, Honda Y, et al. Case reports of portal vein thrombosis and bile duct stenosis after stereotactic body radiation therapy for hepatocellular carcinoma. Hepatol Res. 2014;44:E273-E278.

Reviewer 2 Report

This is a nice review on the subject of SBRT for use in HCC applications. The authors went through published studies exhaustively on different organized aspects, and suggested moving this to early treatment of HCC instead of a substitute therapy as it is currently.

Although many studies are cited, in many situations the results couldn't be compared on equal footing due complex patient composition, diverse status or nature of the disease (HCC), etc. For each situation large prospective studies are needed to draw any conclusions.

A few points to consider:

For example, Table I contains only one study using CPC. What about all the other studies in the Table? Didn't they contain patients classification such as Child-A or -B?

Combined treatment is always expected to be synergistic. maybe the grouping of the surveyed studies would support that more strongly.

SBRT is more accessible than CPT (carbon ion therapy and proton beam therapy). However, the potential of CPT and its spreading use can be a game changer. Please discuss this more.

Author Response

Answers to the reviewers 2’ comments and suggestions

Comments and Suggestions for Authors

This is a nice review on the subject of SBRT for use in HCC applications. The authors went through published studies exhaustively on different organized aspects, and suggested moving this to early treatment of HCC instead of a substitute therapy as it is currently.

Although many studies are cited, in many situations the results couldn't be compared on equal footing due complex patient composition, diverse status or nature of the disease (HCC), etc. For each situation large prospective studies are needed to draw any conclusions.

Answer

Thank you for your comments.

In “2. SBRT in Early-Stage HCC, 2.1. Treatment Outcomes”, “3. SBRT for Portal Vein or Inferior Vena Cava Tumor Thrombi” and “5. SBRT for Extrahepatic Metastasis” part, I have already referred to the necessity of prospective studies.

I add the following sentence in “2. SBRT in Early-Stage HCC, 2.2. Comparison to the Other Treatments” part.

“Further studies, including randomized phase III studies to define which patients are more suitable for each curative local treatment, are needed.”

A few points to consider:

For example, Table I contains only one study using CPC. What about all the other studies in the Table? Didn't they contain patients classification such as Child-A or -B?

Answer

Thank you for your comments.

I added the classification of Child-Pugh class A/B in Table 1.

Combined treatment is always expected to be synergistic. maybe the grouping of the surveyed studies would support that more strongly.

Answer

Thank you for your comments.

I think Table 5, which are summarized combined immunotherapy, shows the expectation of combined treatment with SBRT.

SBRT is more accessible than CPT (carbon ion therapy and proton beam therapy). However, the potential of CPT and its spreading use can be a game changer. Please discuss this more.

Answer

Thank you for your comments.

I have already described the potential of CPT in “2.2. Comparison to the Other Treatments” part. I added the following sentences in Discussion part.

“Third, the segregation of SBRT and CPT has been an important issue. As described in the previous part, there is no doubt that CPT is advantageous due to its physical aspects, especially for large size HCC. On the other hand, for early-stage HCC, although CPT is advantageous from the viewpoint of adverse events from the meta-analysis, LC is no large difference comparing to SBRT at present. Considering accessibility and cost, it seems realistic at present to segregate large HCCs (> 4-5cm) that are difficult to control with SBRT. The NCCN guideline states that "proton beam therapy may be appropriate in specific situations"[8]. To establish evidence, several studies comparing with other local modalities, such as RFA are ongoing, there is possibility that CPT has a higher priority than other modalities in the selection of local treatment for HCC in the near future.”
